# Distress Signals: Age Differences in Psychological Distress before and during the COVID-19 Pandemic

**DOI:** 10.3390/ijerph20043549

**Published:** 2023-02-17

**Authors:** Sandra Hale, Joel Myerson, Michael J Strube, Leonard Green, Amy B. Lewandowski

**Affiliations:** Department of Psychological & Brain Sciences, Washington University in St. Louis, St. Louis, MO 63130, USA

**Keywords:** psychological distress, COVID-19 pandemic, age differences

## Abstract

Psychological distress reached historically high levels in 2020, but why, and why were there pronounced age differences? We address these questions using a relatively novel, multipronged approach, part narrative review and part new data analyses. We first updated previous analyses of national surveys that showed distress was increasing in the US and Australia through 2017 and then re-analyzed data from the UK, comparing periods with and without lockdowns. We also analyzed the effects of age and personality on distress in the US during the pandemic. Results showed distress levels and age differences in distress were still increasing through 2019 in the US, UK, and Australia. The effects of lockdowns in 2020 revealed the roles of social deprivation and fear of infection. Finally, age-related differences in emotional stability accounted for the observed age differences in distress. These findings reveal the limitations of analyses comparing pre-pandemic and pandemic periods without accounting for ongoing trends. They also suggest that differences in personality traits such as emotional stability modulate responses to stressors. This could explain age and individual differences in both increases and decreases in distress in response to changes in the level of stressors such as those occurring prior to and during the COVID-19 pandemic.

## 1. Introduction

“The pandemic certainly was causal, to some degree. But in many respects, it was simply revelatory. It revealed things that were already there, things that have been under the hood, waiting for something to come along and pull the curtain back.”(Curt Thompson)

Quoted in the Tish Harrison Warren newsletter for Times subscribers, What if Burnout Is Less About Work and More About Isolation? New York Times, 9 October 2022. (https://www.nytimes.com/2022/10/09/opinion/burnout-friends-isolation.html, accessed on 9 October 2022).

Psychological distress in the US reached historically high levels after the COVID-19 pandemic began in 2020, and it remains high today. A recent review of changes in mental health associated with the first year of the COVID-19 pandemic summarized the increases in anxiety, depression, and general psychological distress [1]. The overwhelming evidence suggests that the US population is suffering from mental health problems to an unprecedented degree, and as the US Surgeon General noted, these problems are particularly acute for adolescents and young adults [2]. Although we wholeheartedly agree that the situation has reached crisis levels, we would note that the most obvious interpretation, which is that the pandemic stressors caused this crisis, comes with an important caveat. The data appear consistent with the notion that pandemic stressors caused both overall increases in psychological distress and increases in the differences in distress between age groups. We would argue, however, that one must consider more than just the immediately preceding level of distress when interpreting the current levels of psychological distress.

Indeed, there is evidence to suggest that stressors associated with the pandemic may have come on the heels of an ongoing mental health pandemic, potentially creating a “perfect storm” of psychological distress. Twenge et al. [3], for example, reported that in the United States, distress levels on the National Survey of Drug Use and Health (NSDUH) had been progressively increasing in adolescents and adults for some years up to and including 2017, and a similar trend was observed on the biennial Household Income Dynamics in Australia (HILDA) survey over the same time period [4,5]. If that trend continued through 2019, then a further increase from 2019 to 2020 might have occurred even if there had not been a pandemic. We would argue that given a previously increasing trend in psychological distress, subsequent data will need to be compared not just with data from the previous year, but rather with predictions that take existing pre-pandemic trends into account.

Accordingly, the first major goal of the present effort was to update the evidence for those trends, focusing on age differences in distress and how these differences have changed. This is especially important because of the roles played by both age and distress during the COVID-19 pandemic. With respect to age, it is now well established that the risk of severe complications from infection with the SARS-CoV-2 virus is greater for older adults as they are more likely to require hospitalization [6] and also to die as a result of infection [7,8]. With respect to psychological distress, it has been known for some time that distress increases the risk of infections by interfering with the function of the immune system [9,10]. More recently, studies have shown that distress also increases the risk of infection by interfering with mitigation behaviors such as social distancing [11]. Because of these things, it is fortunate that the level of psychological distress experienced by adults decreases with age, as evidenced by the results of national surveys of very large, representative samples such as the aforementioned NSDUH and HILDA.

The strengths of these annual and biennial national surveys are their large samples and their consistent methodology and sampling procedures. This means that they can provide information on long-term trends that provide the context needed for interpreting short-term changes. Although much of the survey data has been published previously, it has often been discussed in terms of specific national implications (e.g., tracking drug use) in keeping with the missions of these surveys. Here we bring their results together to be examined and compared using the same analytic approach. Unfortunately, however, the pandemic led to disruptions in some of the consistency that is one of the greatest strengths of national surveys. Indeed, the NSDUH collected data in only the first and fourth quarters of 2020, and the timing of HILDA, which collects data biennially, was such that no data were collected during the first six months of the pandemic in Australia.

Our second major goal was to examine changes in distress levels throughout the first year of the pandemic. The one national survey that did collect data continuously throughout 2020 was the United Kingdom Household Longitudinal Survey (UKHLS), albeit with significant changes in procedure and smaller samples. Importantly, the UK underwent two national lockdown periods during that year, and thus the UKHLS data, in addition to allowing us to assess the generality of the trends observed in pre-pandemic data from the NSDUH and HILDA, provided a unique opportunity to examine the effects of social deprivation on distress during the pandemic.

Given the pronounced age differences in psychological distress and in the size of the changes in distress levels observed in all three national surveys analyzed here, our third major goal was to examine the extent to which these changes reflected various pandemic stressors and personality traits. Whereas previous efforts to explain age-related differences in psychological distress have tended to focus on why distress has increased so markedly in adolescents and younger adults (e.g., Murthy [2]), the present analyses also address the question of why distress levels have remained so much more stable in older adults. The larger issue here is whether these are really two research questions—first, why were younger adults so vulnerable to the psychological effects of the COVID-19 pandemic, and second, why were older adults so resilient—or should researchers focus on the extent to which a single mechanism underlies the observed age differences in people’s responses to the pandemic? The present analyses took the latter approach and used data collected during the pandemic to examine whether personality traits, particularly traits that change with age (e.g., conscientiousness, emotional stability) might be the basis for such a mechanism.

Thus, our fourth and final goal concerned the implications of age and individual differences in personality traits for interpretation of the national survey data. At issue was the form of a model that might describe the pattern revealed by trend analyses. More specifically, regardless of whether distress increased or decreased over time, the degree to which adults’ distress levels changed was inversely related to their age—the exact opposite of what one might have expected based on age differences in the risk of severe negative consequences from infection with the SARS-CoV-2 virus. Two conceptual models are considered, an additive model and a modulatory model in which trait scores predict the degree to which individuals are affected by external stressors. The pattern in the national survey data is considered in light of the predictions of the additive and modulatory models.

Taken together, the present investigation focused on four questions regarding recent changes in the level of distress. The first question was whether the increasing trend both in psychological distress and in the size of age differences in distress continued right up through 2019, the year before the COVID-19 pandemic began. The second question concerned how much of the distress during the first year of the pandemic role was due to mandated lockdowns and again, whether age differences changed when the overall level of distress increased and decreased over that time period. Finally, the third and fourth questions were whether individual and age differences in psychological distress might be explained by differences in personality traits such as emotional stability, and whether the role of such traits, both before and during the pandemic, was more accurately described by an additive model or one in which traits played a modulatory or interactive role.

## 2. Updated Trend Analyses of Distress in the US and Australia

In order to answer our first question, we began by following up on Twenge et al. [3], analyzing the trend in psychological distress in the US between 2008 and 2019 as revealed in annual reports of the results of the NSDUH. Prior to Twenge et al., several other studies had documented increases in mood disorders and suicide-related outcomes among adolescents since 2010 [12,13,14] and estimated prevalence among college-aged individuals [15,16,17], but little research had examined trends across age groups. Recent trends in older adults in particular had received little or no study.

Next, we followed up on Butterworth et al. [4], analyzing the trends in distress in Australia between 2007 and 2019 using data depicted in the 2021 report of the HILDA. Just as Twenge et al. [3] had highlighted the fact that levels of distress in the US were increasing in the years prior to 2017, Butterworth et al. [4] reported that a similar increase was occurring over the same time period in Australia.

Although the NSDUH and HILDA use very similar questionnaires, the K-6 and the K-10 (which is a longer version of the K-6 [18]), they differ in a number of other respects. The NSDUH is a nationwide survey conducted annually to provide up-to-date information annually on tobacco, alcohol, and drug use, mental health, and other health-related issues in the United States. In contrast, the HILDA is a panel study in which most of the data are collected from the same households. It collects information about economic and personal wellbeing, labor market dynamics, and family life, enabling Australian policy makers to make informed decisions across a range of policy areas, including health, education, and social services.

The distress measures used by the two surveys are similar, but the procedures they follow are quite different. Despite these differences and the fact that the samples came from two different countries, Butterworth et al. [4] reported that distress was also increasing in Australia over the same period that Twenge et al. observed distress increasing in the US [3]. At issue in the present analyses is whether the systematic increases in distress observed through 2017 continued in 2018 and 2019, whether parallel changes occurred in the US and Australia, and whether the substantial differences in procedure were associated with differences in results. The results have implications for assessment of the effects of the pandemic on distress levels: The existence of increasing trends would imply that the level of distress in 2020 might well have been higher than the level in 2019 even if a pandemic had not occurred. Indeed, simple comparisons of the levels of psychological distress in the two years likely overestimate the effect of the pandemic.

### 2.1. Materials and Methods

The NSDUH reports psychological distress data for three adult age groups: participants whose ages when they were tested were 18–25 and 26–49, as well as those 50 and older. Distress was assessed using the Kessler-6 (K-6). As its name suggests, the annual NSDUH reports summarize current levels of drug use and health-related psychological measures. The summary data for each adult age group were reanalyzed here and fitted with quadratic functions whose parameters allow extrapolation of the functions to predict survey responses in 2020.

In contrast to the NSDUH, which surveys a different sample every year, the HILDA is a panel study that contacts members of the same households every other year. Like the NSDUH, which has used the K-6 in its present form to assess psychological distress since 2008, the Hilda has used the K-10 to assess psychological distress since 2007. The two scales differ in that the K-6 is a six-item questionnaire with four items related to depressive symptoms (e.g., hopelessness) and two related to anxiety symptoms (e.g., restlessness) whereas the K-10 adds another question related to anxiety and three more related to depression.

Although the published analyses of the HILDA provide data for six age groups, for the present purposes these data were averaged to yield three age groups similar to those used by NSDUH to facilitate comparison. Unfortunately, the HILDA survey did not conduct interviews from the beginning of February until August in 2020—the first six months of the pandemic in Australia—irrevocably complicating any efforts to compare observed and predicted distress levels to assess the effects of the pandemic.

### 2.2. Results

Figure 1 replots data from three national surveys, the NSDUH, HILDA, and UKHLS, in order to illustrate the similarity of the increasing trends in psychological distress in the U.S., Australia, and the U.K., respectively, over the decade preceding the COVID-19 pandemic. Panel a presents the percentage of participants in each of the NSDUH’s three age groups who reported experiencing serious psychological distress in the preceding month. Consistent with Mojtabai and Jorm [19], distress levels were relatively stable for all three age groups until around 2013, although levels decreased with the age of the group. Subsequently, however, distress in the two younger groups began to increase in a positively accelerated fashion so that their data from 2008 to 2019 was well described by quadratic functions fit to the percentages. More specifically, the *R*^2^s for the NSDUH percentages from the 18–25-year-old and 26–49-year-old groups were 0.985 and 0.892, respectively. Moreover, linear fits to the percentages from the last five years before the pandemic (2015–2019) revealed significant increases in distress for both groups (both *p*-values < 0.005). Notably, the annual rate of increase for the 18–25-year-old group was 1.29 percentage points per year and the rate for the 26–49-year-old group was only 0.44 points annually, whereas no significant change in psychological distress was observed in those 50 years of age and older (*p* > 0.35), and the *R*^2^ was only 0.285 (see Table 1, which illustrates the quantitative similarities in the increasing trends in psychological distress for different age groups on the NSDUH and HILDA).

After fitting the NSDUH distress data from 2008 to 2019, we used the parameters of the best-fitting quadratic function for each age group to estimate what that group’s distress level would have been in 2020 if a pandemic had not occurred (see the dotted prediction lines in panel a of Figure 1). Although these predictions may be compared visually with the distress levels that were actually observed, difficulties collecting data during the COVID-19 pandemic required major procedural changes and may have affected sample characteristics, leading the NSDUH to warn against conducting actual statistical tests or at least to interpret their results very cautiously. Prior to 2020, for example, data were collected throughout the year using in-person interviews, whereas in 2020, data were collected only during the first and fourth quarters, and in the fourth quarter, most data were collected online using self-administered interviews, resulting in a sample that was more highly educated than in previous years.

Nonetheless, the analysis of the NSDUH data depicted in panel a of Figure 1 provides a clear illustration of what we believe is the appropriate statistical approach to comparing psychological distress in 2020 to that in previous years. That is, when it is possible to do so, researchers should assess changes by extrapolating the trend prior to the pandemic and then compare the resulting extrapolated values to observed values, rather than simply comparing the pandemic year to the preceding year. The focus in Figure 1, however, is really on the changes in psychological distress prior to the pandemic and the age differences in these changes. As may be seen, the percentage of young adults (ages 18–25) reporting severe psychological distress approximately doubled from 2008 to 2019 whereas the increase in the percentage of middle-aged adults (ages 26–49), while significant, was much smaller. In contrast, distress in the older adult group (ages 50 and up), did not increase significantly.

As may be seen in panel b of Figure 1, the pattern in the Australian HILDA data is similar to that observed in the US NSDUH data—an initial period of stability with the highest level of distress observed in the youngest group followed by a period of increase, with the rate of increase decreasing with age and baseline level. As with the analyses of the NSDUH data, the fits of quadratic functions to the HILDA data from the two youngest groups were again excellent, with *R*^2^s of 0.897 and 0.970 for the 15–24-year-old and 25–49-year-old groups, respectively. Moreover, the fit of a quadratic function to the data from those 55 and older was also very good: *R*^2^ = 0.849.

To better assess the increases in distress in the years preceding the COVID-19 pandemic, linear regression analyses were conducted of the NSDUH and HILDA data for each age group from the last five waves studied prior to 2020 (see Table 1). Similar results were obtained for both surveys. Linear regression models with significant slopes accounted for more than 90% of the variance in distress for the two younger groups. In contrast, the slopes were not significant for the oldest group in either the US or Australia samples. Not only did the slope for the youngest group differ significantly from that for the middle age group, but the slope for the youngest group was more than twice that for the middle age group.

### 2.3. Discussion

Taken together, these results indicate that the increasing trends in psychological distress in both the US and Australia first observed by both Twenge et al. and Butterworth et al. [3,4] continued at least right up to the beginning of the COVID-19 pandemic. This was clearly true for all but the oldest age groups, and the trends were significantly stronger for the youngest groups than for those from the middle age groups. The similarity of the results obtained in both countries despite all the methodological differences between the NSDUH and the HILDA (e.g., cross-sectional versus panel) survey designs testify to the robustness of the phenomena. The present results also appear to rule out the possibility that the causes of the observed increases in psychological distress are specific to one nation, although clearly these increases could be specific to economically well-developed English-speaking nations.

The increasing trends prior to the beginning of the pandemic are important methodologically because they raise the possibility that comparisons that do not take these increasing trends into account overestimate the deleterious effects of the pandemic on psychological wellbeing. They are also important from a public health perspective because they suggest that there was an ongoing mental health crisis prior to the pandemic caused by the SARS-CoV-2 virus. The onset of the COVID-19 pandemic distracted from what may have been the beginning of a mental health crisis and focused attention on the contribution of the virus to people’s distress. Should the virus be controlled in the future, we may well be left with this pre-existing crisis. Understanding its origins could be critical to dealing with it effectively.

## 3. Lockdowns, Quarantines, and Stay-at-Home Orders in the UK

Of all the studies examining the mental health effects of the pandemic reviewed by Aknin et al. [1], only one tracked psychological distress across the years before the pandemic for purposes of comparison with distress in 2020. Specifically, Pierce et al. [20] examined data from the annual UK Household Longitudinal Study (UKHLS), a methodologically consistent panel survey given annually to tens of thousands of participants in Great Britain. They reported that distress in 2020 was initially greater than what would be predicted based on pre-pandemic trends, but Daly and Robinson [21] subsequently reported that distress, as measured by the UKHLS, returned to pre-pandemic levels before the year was out. In contrast, our own analyses of the UKHLS data reported here paint a more complicated picture of distress in the UK during the pandemic and suggest that national lockdowns or stay-at-home orders, with their attendant social deprivation, have played a role at least as critical as that of health concerns (e.g., estimated chance of infection) in generating psychological distress.

As highlighted in this second part of our investigation, there were two national, government-mandated lockdowns in the UK during the first year of the pandemic, and thus the UKHLS data from 2020 provide information about both the effects of imposing a lockdown and the effects of terminating a lockdown. Although it is clear from the literature that quarantines have predominantly negative, possibly long-lasting effects on mental health [22,23], the circumstances in 2020 were somewhat unusual, coming as they were during what appears to us to have been an ongoing mental health crisis. Indeed, for current purposes, the increasing trend in psychological distress prior to the COVID-19 pandemic is not just noise to be statistically controlled in order to better understand the impact of COVID-19. Rather, this trend sheds light on the mental health pandemic prior to 2020, as well as providing further evidence of the international character of the increases in psychological distress and evidence against purely national causes of these increases.

### 3.1. Materials and Methods

The UK Household Longitudinal Study (UKHLS) is an ongoing panel survey that, prior to the COVID-19 pandemic, collected data from more than 40,000 households using mainly face-to-face interviews. Rather than basing analyses on the percentages of participants meeting specific distress criteria such as those reported in the periodic, publicly available reports of the NSDUH and HILDA surveys, the analyses of the UKHLS reported here were based on raw data available from the UK Data Service repository. Having access to the raw data made it possible to construct age groups in exactly the same way as that done with the NSDUH and to calculate and analyze the means for each age group in the same way.

The data from the 2008–2019 period were collected following the usual UKHLS procedures. During the last week in April of 2020, the first year of the pandemic, members of households that had participated in either of the two most recent waves of UKHLS data collection were invited to complete the first wave of the COVID-19 web survey if they were 16 or older. They also were asked to complete follow-up surveys online, initially every month and then every other month throughout 2020.

As in previous years, the 12-item General Health Questionnaire (GHQ-12) was used to measure psychological distress in 2020. Data were analyzed here using the actual scores for each participant, coded in the way recommended by the developers of the test, and using the same age groups as with the NSDUH in order to facilitate comparisons of the data from the US and UK. It should be noted, however, that testing for the UKHLS was undertaken online rather than face-to-face. Moreover, sample sizes in 2020 were less than half of those in 2019 and varied considerably both between age groups and from wave to wave. As with the NSDUH, the considerable differences in procedure between 2019 and 2020 suggest that statistical comparisons may be inappropriate.

### 3.2. Results

Examination of Figure 1 reveals the similarity of the patterns of increase in psychological distress on the UKHLS (panel c) and both the NSDUH and the HILDA. As may be seen, distress increased the most in those aged 18–25 while smaller increases were observed in the middle age group (ages 26–49), whereas distress did not increase at all in the older adult group (age 50 and greater). In addition, as with the NSDUH and HILDA, linear regression analyses were conducted for each age group using data from the last five waves studied prior to 2020 (see Table 2). Similar results were obtained for all three surveys. Linear regression models with significant slopes accounted for more than 90% of the variance in distress for the two younger groups. In contrast, the slopes were not significant for the oldest group in either the US or Australia. Not only did the slope for the youngest group differ significantly from that for the middle age group, but the slope for the youngest group was more than twice that for the middle age group.

Finally, because measures of distress on the GHQ-12 were available from both before and after the pandemic began, we calculated what the level of distress would have been in the absence of a pandemic if the increasing trend simply continued from 2019 to 2020. As with the NSDUH extrapolations, this was done using the parameter estimates for each age groups’ best-fitting quadratic function (also depicted in Figure 1). As was the case with the NSDUH, however, the validity of statistical comparisons of observed and predicted distress levels clearly would be open to question because of the major procedural differences in sampling and administration of the UKHLS before and during the COVID-19 pandemic. Nonetheless, the predicted values may provide useful, albeit informal, benchmarks, and accordingly they are included in Figure 2, which presents levels of distress on the UKHLS during the first year of the pandemic in the United Kingdom. Importantly, the 2020 data show that although distress increased when a lockdown was imposed and decreased when it ended, younger groups consistently showed larger changes, both larger increases and larger decreases, than older groups.

### 3.3. Discussion

#### 3.3.1. Pre-Pandemic Trends

The UKHLS data provide further evidence of the generality and robustness of the current findings regarding psychological distress prior to the COVID-19 pandemic. Analysis of the UKHLS, NSDUH, and HILDA surveys revealed similar patterns of results obtained regardless of whether the data collection used cross-sectional (NSDUH) or longitudinal/panel (UKHLS and HILDA) survey designs, and regardless of whether they used the shorter (NSDUH) or longer (HILDA) versions of the Kessler distress questionnaire or a completed different questionnaire, the GHQ-12 (UKHLS). Similar trends were also observed regardless of whether the data analyzed here were reported in terms of the percentages of participants who did or did not exceed some threshold for serious or clinically relevant distress or whether, as in the case of the UKHLS, the raw questionnaire data were analyzed.

Despite all these differences, including different samples of participants from three different countries, the data from all waves collected since 2007 or 2008 are well described by quadratic trends, with little or no change up until about 2015, followed by almost linear increases from around 2015 to 2019, the year before the pandemic. Finally, regression analyses of the data from the last five waves prior to the pandemic revealed that not only were younger participants consistently more distressed than middle-aged ones, but in each case, psychological distress increased at more than twice the rate for the younger groups compared to the older ones.

Taken together, these results suggest that even though there is still no consensus as to what had been causing distress levels to increase prior to the pandemic, whatever the stressors were, people’s sensitivity to them clearly decreased with age. Importantly, although adolescents and young adults may have been the most sensitive, distress in middle-aged and older adults appears to have increased roughly in synchrony, albeit not to the same extent.

In order to facilitate comparisons of the US with Australia and the UK, we used age groupings similar to the NSDUH groups in our analyses. However, these groupings obscured evidence of pre-pandemic increases in psychological distress in younger older adults. In this regard, it should be noted that the percentage of Britons aged 55–69 whose distress was “clinically significant” on the UKHLS increased progressively from 14.8% to 17.0% between 2014–2015 and 2018–2019 and the percentage of Australians aged 55–64 who reported “serious psychological distress” on the HILDA increased from 14.9% in 2015 to 18.5% in 2019. In addition to providing comparative information on pre-pandemic trends in psychological distress, we believe that these age-related differences in distress, taken together with the age-related differences in the rates at which distress increased, are major clues as to the mechanism(s) underlying age and individual differences in people’s responses to stressors under many circumstances, and not just when people’s health and well-being is threatened by a pandemic. Importantly, these results suggest that although it may be extremely useful to group participants for purposes of statistical analyses, there may in fact be one or more age-sensitive traits, and that looking for the causes of distress in different age groups separately may preclude discovering the source(s) of an underlying continuous variation in sensitivity.

#### 3.3.2. Lockdowns, Quarantines, and Stay-at-Home Orders in the UK

Pierce et al. [20] and Niedzwiedz et al. [24] appropriately compared psychological distress in the UK at the beginning of the pandemic (April 2020) with the level of distress predicted by previous upward trends. They found that although distress increased from 2015 to 2019, a significantly larger increase was seen from 2019 to April 2020, with younger participants showing the largest increases. However, as shown by the present analyses of repeated testing during the pandemic that we conducted to answer our second question, their finding, while accurate, was only part of the story.

A major strength of the NSDUH, HILDA, and UKHLS has been each survey’s consistent methodology. In order to examine the time course of the pandemic in greater detail, however, the UKHLS asked for volunteers to undergo repeated online testing, initially once per month and then changing to once every two months, and then even less frequently. The size of the resulting sample was approximately half that of the usual sample, and that fact, taken together with the special recruitment, the online testing, and the frequently repeated tests, are the kinds of procedural changes that call for caution in interpreting the results of statistical comparisons.

Visual comparisons, nevertheless, provide important information regarding the effects of the pandemic. For example, the timing of observed changes in distress in 2020 coincided with the beginning and ending of periods of government-ordered lockdowns in the UK [25]. The first lockdown began in the last week of March 2020 and continued for approximately three months (through June), at which time the prime minister announced that the period of “national hibernation” was ending, and distress levels decreased for all three age groups. As may be seen in Figure 2, the older the group, the smaller the size of the decrease in distress. A second official “hibernation” period, however, came into effect in the first week of November and continued for approximately one month, resulting in increases in distress in all three age groups. It should be noted that the younger the group, the larger the size of these increases. Taken together, these findings suggest that regardless of their direction, the size of changes in psychological distress are generally age-dependent, with sensitivity to stressors decreasing with age. In between lockdowns, distress levels were close to those predicted from pre-pandemic trends, as represented by the dotted lines in Figure 2, which are the values for the trend curves shown in the bottom panel of Figure 1 extrapolated to 2020. Overall, the results shown in Figure 2 suggest that most of the increase in distress in the UK during 2020 is attributable to characteristics of the lockdown periods, including social deprivation, and not to the COVID-19 pandemic per se.

Further evidence of the effects of lockdowns on psychological distress comes from Australia where a surge in infections in the state of Victoria led to a lockdown in that state alone. Notably, equivalent levels of distress were observed in Victoria and other states prior to the lockdown, suggesting that the different states did not differ appreciably with respect to ongoing trends in psychological distress. During the period when lockdown was in effect in Victoria, however, symptoms of both anxiety and depression were significantly higher than in other states [26,27]. In addition, a national lockdown was imposed in New Zealand in 2020. Sibley et al. [28] compared propensity-score matched groups measured before and during the first 18 days of lockdown and found that distress as measured by the K-6 increased significantly; similar results were observed when individuals’ scores from the year before the lockdown were compared with their scores during the first 18 days of the lockdown.

It may be recalled that, in the US, most states also had issued stay-at-home orders by the beginning of April 2020, the point at which distress peaked in the US according to a study by Daly and Robinson [21]. These researchers observed that distress decreased rapidly, returning to baseline by the middle of May, by which time all states had at least started lifting their stay-at-home orders. Interpretation is complicated by the fact that different states issued their orders and lifted them at different times. Moreover, the longitudinal nature of the study (participants were scheduled to be tested every two weeks) raises the possibility of testing effects. Nevertheless, the pattern, which is similar to that observed in the UK and also in Germany [29], has important theoretical implications to be considered in the following section.

Although the increasing trends in psychological distress from the three national surveys just reviewed suggest that studies that simply compare distress levels prior to and during the first year of the COVID-19 pandemic likely overestimate the effects of the pandemic, that does not mean that events associated with the pandemic did not lead to significant increases in anxiety and depression. Rather, changes in distress levels during the pandemic, notably those associated with the lockdowns in the UK, Australia, New Zealand, and Germany [29], suggest that the questions asked by many previous studies have been too broad. That is, the COVID-19 pandemic is a complex sequence of events. It affects different groups of people differently (e.g., young vs. old; women vs. men), and it even affects the same group differently at different points in time. Moreover, the disease, the virus causing it, and the political and public health responses to the pandemic all have changed since the pandemic began and appear likely to keep changing.

## 4. Age and Individual Differences in Psychological Distress

We recently hypothesized that social deprivation has been a major cause of psychological distress during the COVID-19 pandemic [11,30]. We hypothesized further that if social distancing increased feelings of isolation, it could lead distressed individuals to avoid engaging in distancing and exacerbating their distress. Moreover, we reasoned that because vaccination offered the promise of a return to previous levels of socialization, distress might actually increase the likelihood of vaccination. Indeed, our recent findings are consistent with the hypothesis that distress does have differential effects on vaccination and mitigation behaviors such as social distancing. Specifically, distress levels were negatively correlated with distancing but positively correlated with vaccination, although the strength of these relations, like the degree of distress, decreased with age [11].

These findings, taken together with the observed effects of lockdowns, strongly support the fundamental assumption of the *differential distress hypothesis*: Social deprivation is a major source of the psychological distress associated with the pandemic [11,30]. The findings concerning the UK lockdowns, in particular, raise the question of why people of different ages and genders differ in their sensitivity to such deprivation. Accordingly, we now focus on that question, beginning with analyses of unpublished personality data from our recent study of mitigation and vaccination decisions during the pandemic.

Consistent with our hypothesis regarding the social source of distress during the COVID-19 pandemic, we found previously that loneliness (as measured by the Three-Item Brief Loneliness Scale [31]) was correlated (*r* = 0.528, *p* < 0.001) with distress (as measured by the Hospital Anxiety and Depression Scale [32]. Multiple regression analyses were used to assess the relative contributions of loneliness and other pandemic-related concerns (i.e., community-related and personal concerns about the consequences of the pandemic, participants’ estimates of the likelihood of infection, and the number of acquaintances hospitalized [11]). A reduced model with only the two significant predictors (loneliness and estimated chance of infection) accounted for 43.4% of the variance in distress. Both of these variables are likely to increase during a lockdown, which necessarily involves social deprivation, and this is initiated because of an increased risk of infection, a fact that is usually announced to the public. This suggests that the reduced model could account for both the observed increases in distress during lockdowns and the decreases in distress when they end.

Although we had collected three IPIP NEO personality measures from our sample of over 800 participants, two of which (Conscientiousness and Emotional Stability) increased significantly with age, our focus at the time was on social deprivation and associated consequences of psychological distress rather than on its causes [11]. As a result, we had not fully analyzed the personality measures until the current effort. Now, however, we wondered if age-related differences in personality traits might underlie the decrease in psychological distress with advancing age. After all, although an *R*^2^ of 0.434 indicates that the model is a good fit to the data, the model provides little insight into why distress is negatively correlated with age. In contrast, the analyses in this part of our investigation, the third, which we conducted to address our final pair of questions, do appear to provide such insight

### 4.1. Materials and Methods

As described in detail in Myerson et al. [11], 852 MTurk workers completed an online survey between 16 April to 1 May 2021. As noted in the original published article [11], the survey was approved by the Institutional Review Board of Washington University in St. Louis. The survey consisted of three parts, beginning with questions about mitigation and mask wearing. The second part of the survey consisted of the Hospital Anxiety and Depression Scale (HADS [32], brief Loneliness Scale questions [31], and questions regarding personal connections with COVID-19 cases and subjective opinions about COVID-19 vaccines. The third part of the survey consisted of questions from the IPIP-NEO (International Personality Item Pool-Neuroticism, Extraversion, Openness) personality test, plus questions regarding political affiliations and their 2020 Presidential vote, followed by demographic questions.

### 4.2. Results

One might expect that Emotional Stability (the flip side of Neuroticism) would predict psychological distress in the current situation because it measures both reactivity to negative events and how prone a person is to negative emotional states such as anxiety, depression, and anger (for reviews, see [33,34]). Consistent with this expectation, when Stability was added to the reduced model (see Table 3), the model accounted for 64.9% of the variance in psychological distress and the AICc (i.e., the Akaike Information Criterion with a correction for small sample sizes) decreased from 2896.0 to 2515.4. In contrast, when Conscientiousness was added to the reduced model instead of Stability, the model accounted for 53.8% and the AICc was 2737.2.

Moreover, when both Age and Stability were included in the model, the coefficient for Age was not significant (Table 4). Taken together, these results suggest that age-related differences in distress are better explained by age-related differences in emotional stability than by age-related differences in conscientiousness.

### 4.3. Discussion

The AICc is particularly useful in situations like the present one where the models to be compared are not nested. In the present case, the AICc values and *R*^2^s indicate that a model that includes Stability as a predictor of distress is clearly better than an analogous one that instead includes Conscientiousness as a predictor. Indeed, a similar protective effect of Stability was observed in Germany during the pandemic [29]. It would be of no surprise, of course, if Stability were to modulate responses to stressors in general. Moreover, it would explain why, if stressors increased in the years prior to the pandemic, younger groups showed larger increases in psychological distress than older groups (see Figure 1). After all, an extensive body of research suggests that younger adults have lower Stability scores than older adults (for a review, see [35]).

To say that Stability affects psychological distress, however, does not address the question of how it affects distress. Consider, for example, two conceptual models of the effect of an external stressor on an individual’s level of psychological distress, an additive model and what we have termed a modulatory model:**Additive Model:** Distress = b0 + b1 ∗ Stressor + b2 ∗ Stability
**Modulatory Model:** Distress = b0 + b1 ∗ Stressor/Stability

Now imagine that we measured a person’s distress at two points in time, one before and the other after a change in the magnitude of the stressor. Assuming that the initial magnitude of the stressor was not zero, both models predict that the individual’s initial level of psychological distress would depend on their emotional stability, with more stable individuals being less distressed than less stable individuals. The models differ, however, in the effect of a change in the stressor. According to the additive model, the degree to which psychological distress will change when the stressor changes will be independent of a person’s emotional stability. In contrast, the modulatory model predicts that the degree to which distress will change is inversely related to the emotional stability of the individual. More stable individuals will change less, and less stable individuals will change more. Importantly, this prediction holds regardless of whether the stressor increases or decreases in magnitude: If the stressor increases, the distress of more stable individuals will increase less than the distress felt by less stable individuals, whereas, if the stressor decreases, more stable individuals will show smaller decreases in distress than less stable individuals.

Notably, the effects of lockdowns on the level of psychological distress in the UK during the first year of the COVID-19 pandemic are consistent with the predictions of the modulatory model (see Figure 2), as are the changes in distress in the first few months in the US [21] and during the lockdown in Victoria, Australia [26]. Given that emotional stability increases with age in adults, the modulatory model predicts that when the magnitude of a stressor changes, the size of the resulting changes in psychological distress will vary depending on a person’s age.

Both the additive model and the modulatory model predict that increases in a stressor such as the social deprivation caused by a lockdown will increase distress and that those who are less emotionally stable will experience greater distress at all levels of Stress. However, the additive model predicts that psychological distress will increase to the same extent regardless of a person’s or group’s Emotional Stability, whereas the modulatory model predicts that lower Stability will be associated with greater increases in distress.

It should be recalled that, in the UK, younger individuals showed larger decreases on average than older individuals when the first lockdown there ended and again showed larger increases when the second lockdown began. Because younger adults are known to be less emotionally stable than older adults, these results are clearly more consistent with the modulatory model than with the additive model. The fact that the initiation and termination of lockdowns were in response to increases and decreases, respectively, in the risk of COVID-19 infection, does not change the predictions or weaken the evidence for the modulatory model, although it does make it difficult to separate the effects of social deprivation and concerns regarding the likelihood of infection.

We would note, moreover, that gender differences also provide a test of the modulatory hypothesis. Women are known to score lower on Emotional Stability and higher on measures of both depression and anxiety. Thus, like younger adults, women would be predicted to show larger increases in both depression and anxiety when lockdowns go into effect, and in fact this was the case for the UKHLS pandemic data reported by Pierce et al. [20] and the Australian lockdown data reported by Fisher et al. [26]. Notably, the percentage of women in the UKHLS sample with clinically significant distress increased from 23.0% in 2018–2019 to 33.3% in April 2020, whereas for men, the corresponding percentages were 14.5% and 20.4%, a much smaller change.

Although we would argue that the increases during lockdowns also reflected ongoing trends, the data are clearly consistent with a modulatory role for Stability. With respect to the ongoing trends, moreover, according to the UKHLS data reported by Pierce et al. [20], females showed larger increases in distress than males (3.6% vs. 0.8%) from 2014–2015 to 2018–2019 (i.e., from the first to the last wave prior to the COVID-19 pandemic) who they analyzed. With respect to ongoing trends, examination of the adult data (collapsed across age) from the NSDUH shows that females reported greater psychological distress than males throughout the years before the pandemic. Moreover, as unspecified stressors increased over the pre-pandemic years, females’ distress increased to a greater extent than that of males (Figure 3). Thus, we would argue that both the adult data from the NSDUH as well as the data from UKHLS are consistent with the hypothesis that personality traits such as Emotional Stability modulate the extent to which people respond to changes in stressors as a function of their current level of psychological distress, although other factors certainly may contribute.

Some stressors, of course, may have greater effects than others, and Stability may play a larger role in modulating the contribution of some stressors than it does with respect to other stressors. Moreover, although we have focused on Emotional Stability here because its definition implies a modulatory role, other measures (e.g., Conscientiousness) that change with age could, in principle, also play a modulatory role.

## 5. Conclusions

We posed four questions at the outset of this paper. First, we asked whether the increasing trend both in psychological distress and in the size of age differences in distress continued right up through 2019, the year before the COVID-19 pandemic began. We answered this question by updating national survey data. After updating the data, analyses clearly revealed that the COVID-19 pandemic was preceded by a period of increasing psychological distress in the U.S., Australia, and the U.K., and that younger adults were the most affected and older adults the least. Notably, although young adults showed the largest increases in distress, increases were observed in adults of all ages.

Second, we asked how much of the distress during the first year of the pandemic was due to mandated lockdowns and again, whether age differences changed when the overall level of distress increased and decreased over that time period. Although a precise, quantitative characterization is not possible because of procedural changes in these national surveys, we were able to address our second question based on evidence showing that lockdowns were responsible for increased distress in the U.S, Australia, and the U.K.

The data from the U.K. provide the most detailed picture and indicate that younger, middle-aged, and older adults were all affected, but that the size of changes, both increases when a lockdown was imposed and decreases when it was lifted, were, again, largest for the younger adults and smallest for the older adults. We found that when the first U.K. lockdown was lifted, the level of distress level in each age group decreased to approximately the level predicted based on the increasing pre-pandemic trend. These results raise the possibility that most, if not all, of the increase in distress at the beginning of 2020 was due to social deprivation, but the fact that the lockdown was due to the high rates of infection with the SARS-CoV-2 precludes a definitive test of this possibility.

Our third and fourth questions concerned whether individual and age differences in psychological distress might be explained by differences in personality traits such as Emotional Stability. Our answer to these questions is based on the entire pattern of individual and age-related differences analyzed here. Specifically, we postulate that individuals’ sensitivity to stressors decreases with age, and that this sensitivity modulates the size of changes in distress as the level of stressors changes. By definition, the psychological trait of Emotional Stability is the kind of individual attribute that could play such a modulating role because it increases with age, suggesting that stressors would have less and less of an effect as one gets older. Thus, it predicts both the pattern of age differences in distress levels as well as the age differences in the rates of change in distress that were observed both before and during the pandemic.

Emotional Stability also shows significant gender differences that are consistent with the gender differences in distress levels and in the size of changes in distress observed in our analyses. Notably, however, some other personality traits (e.g., conscientiousness) also show both age and gender differences and thus represent possible alternative modulating factors, although of course, more than one trait may be playing a role here. Further analyses and more data will be needed, but the approach described here appears to us to be a promising one.

Currently, social deprivation appears to play an important role not only in distress levels but also in mitigation behaviors and vaccination decisions, interfering with the former while facilitating the latter [11,30]. Still, pre-pandemic distress levels suggest that if COVID-19 were to become endemic rather than pandemic, distress levels most likely would still be relatively high, and would remain a public health problem. Not only does psychological distress decrease one’s quality of life, but it also has adverse health effects, including decreased immune function (e.g., [9,10]). In fact, distress appears not only to increase the likelihood of hospitalization from COVID-19 (e.g., [36]) but also the likelihood of long-COVID symptoms [37]. Thus, high distress levels potentially pose a double danger, as distress may increase the likelihood of exposure to the SARS-CoV-2 virus by interfering with mitigation behaviors while also decreasing the body’s ability to fight infection during the pandemic.

All this, moreover, comes on the heels of increases in distress indicative of an international mental health pandemic that preceded the COVID-19 pandemic, at least in English-speaking countries. Taken together, the pre-pandemic increases followed by those associated with the pandemic may have created a “perfect storm” of psychological distress, at least during mandated lockdowns. As a result, there is clearly a lot at stake in estimating levels of psychological distress and the extent to which changes reflect the current pandemic, as well as in the ability to predict future distress levels.

The present findings highlight the extent to which on-going trends must be considered as well as the likelihood that failure to do so may lead to overestimating the effect of the pandemic, per se. Still, it is likely that even after taking ongoing trends into account, distress levels really did increase somewhat because of the pandemic [20,38], in large part because of social deprivation. Our analysis of the UKHLS data, for example, showed that increases in distress came and went with the onset and offset of government-imposed lockdowns. This finding reinforces the idea that social deprivation was a key determinant of increases in psychological distress during the pandemic. Failures to engage in social distancing and other mitigation behaviors may be one manifestation of this idea [11,30]. Moreover, these failures could result in a positive feedback loop, as less mitigation could result in more infections, which in turn could lead to lockdowns and even greater distress. Fortunately, there is an extensive literature reviewed in [39] that provides suggestions regarding individual-, organizational-, community-, and even nation-level strategies that may aid in managing distress and promoting resilience if they are utilized effectively.

Finally, our analysis of previously unreported personality measures from the recent study by Myerson et al. [11] suggests that traits such as emotional stability that change with age and differ between men and women can predict individual differences in psychological distress at a given point in time. Moreover, differences in such personality measures can also predict individual differences in the extent to which changes in external stressors produce changes in psychological distress. Although the modulatory model described here oversimplifies the situation and will require more rigorous testing, we believe it represents a step forward in our understanding of the determinants of psychological distress, not only during the COVID-19 pandemic, but also before and even after the pandemic. The model raises the possibility that, in future health crises, older adults could be both those at greatest risk and those who are hardest to motivate because they are also likely to be the least distressed. Such possibilities highlight the need for research that addresses the problems inherent in messaging people of all ages when, as the modulatory model suggests, one size may simply not fit all.

## Figures and Tables

**Figure 1 ijerph-20-03549-f001:**
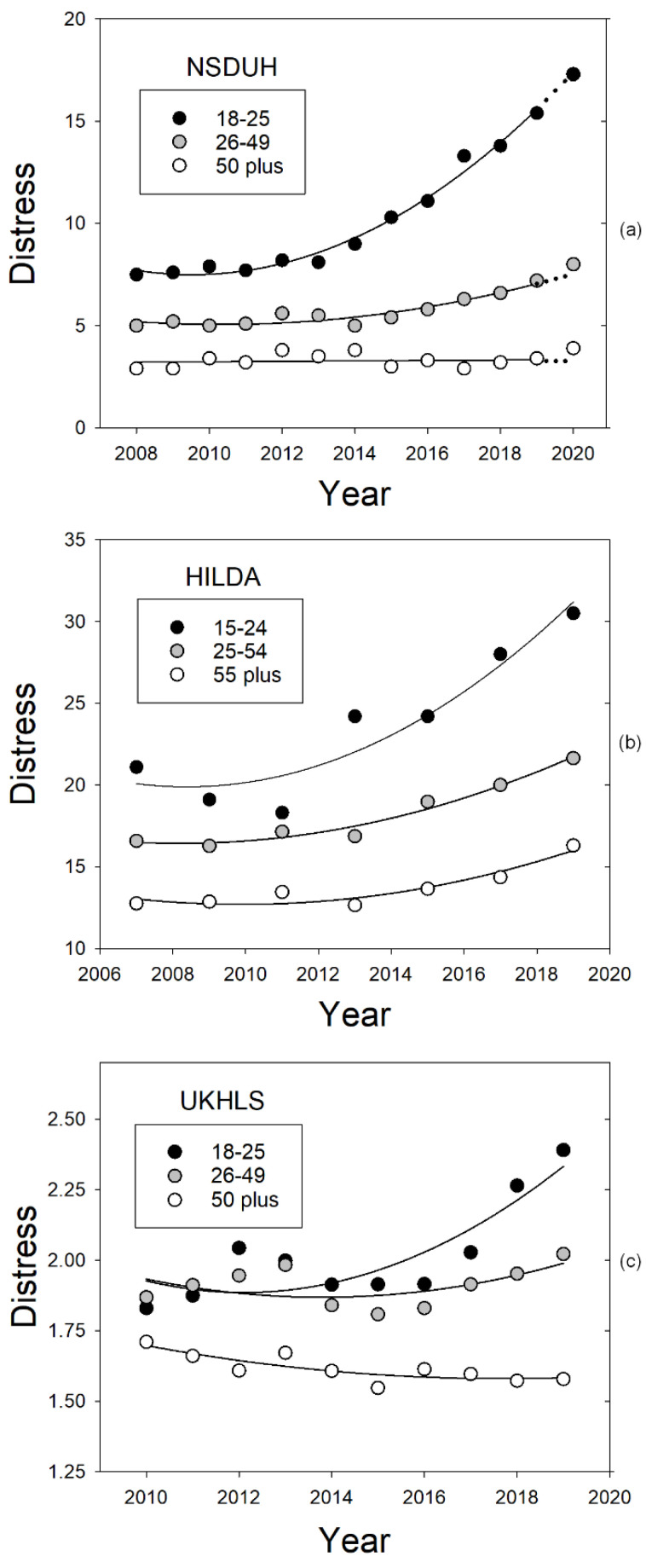
Changes in psychological distress across the years prior to the COVID-19 pandemic. Solid curves represent the quadratic functions that best fit the data through 2019 for the three age groups in each panel. Note that both axes vary slightly depending on the survey and that, whereas panel (**a**) and panel (**b**) present percentages experiencing serious distress, panel (**c**) presents mean scores on the GHQ-12. The dotted lines shown in the upper panel represent the predictions of the functions that best described distress in the US from 2008 to 2019 for each age group. (The predictions for the UK based on their national survey, the UKHLS, are not depicted here).

**Figure 2 ijerph-20-03549-f002:**
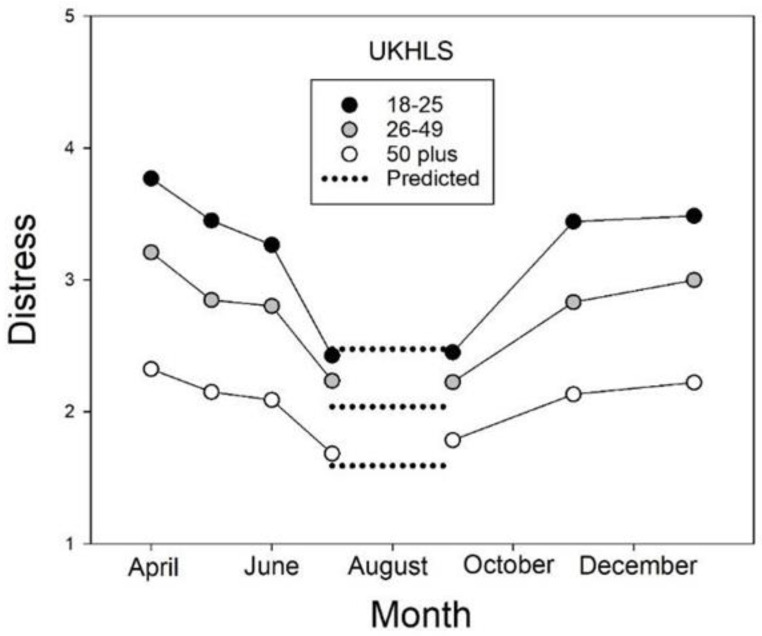
Psychological distress as a function of month in 2020 from the UK Household Longitudinal Study (UKHLS). Note that the first lockdown was in effect in April through June and the second lockdown began in November.

**Figure 3 ijerph-20-03549-f003:**
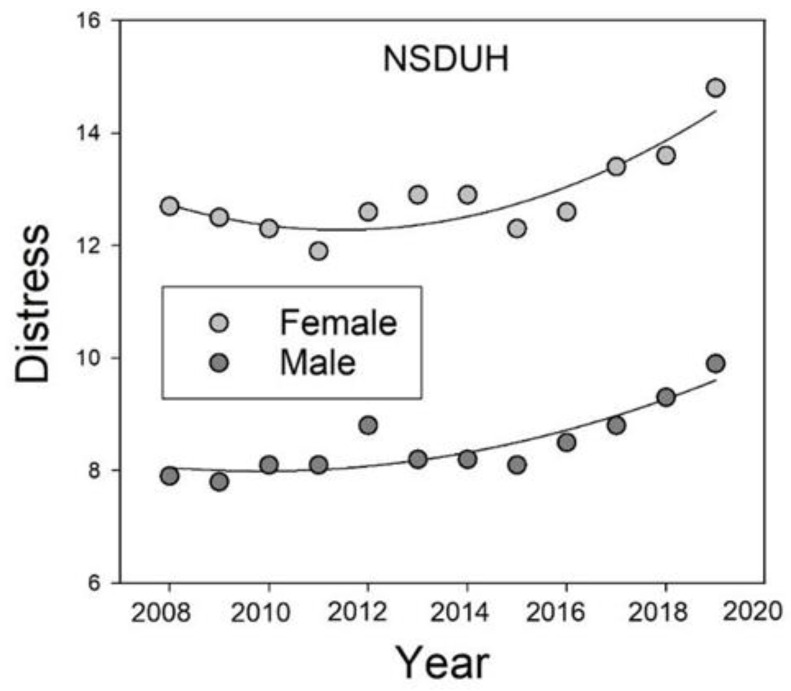
Percentages of females and males reporting serious psychological distress in the past year on the NSDUH surveys from 2008 to 2019.

**Table 1 ijerph-20-03549-t001:** Regression analyses (coefficients, standard errors, *t*, and *p*-values) of the last five waves of the National Survey of Drug Abuse and Health (NSDUH) and the Household Income Dynamics in Australia (HILDA) before 2020.

Survey	Ages	*R* ^2^	Coeff.		Std. Err.	*t*	*p*
NSDUH							
	18–25	0.970	Int ^1^	−9.150	2.218	4.126	0.026
			Slp ^2^	1.290	0.130	9.923	0.002
	26–49	0.992	Int	1.220	0.394	−3.097	0.053
			Slp	0.440	0.023	19.053	<0.001
	50+	0.284	Int	1.970	1.092	1.803	0.169
			Slp	0.070	0.064	1.093	0.354
HILDA							
	15–24	0.931	Int	3.890	3.381	1.151	0.333
			Slp	1.410	0.221	6.365	0.008
	25–54	0.924	Int	9.820	1.537	6.387	0.008
			Slp	0.607	0.101	6.023	0.009
	55+	0.718	Int	8.530	2.044	4.173	0.025
			Slp	0.370	0.134	2.763	0.070

^1, 2^ Int = Intercept, Slp = Slope.

**Table 2 ijerph-20-03549-t002:** Regression analyses (coefficients, standard errors, *t*, and *p* values) of the last five waves of the UK Household Longitudinal Study (UKHLS) before 2020.

Survey	Ages	*R* ^2^	Coeff.		Std. Err.	*t*	*p*
UKHLS							
	18–25	0.914	Int ^1^	−0.110	0.392	−0.280	0.798
			Slp ^2^	0.130	0.023	5.657	0.011
	26–49	0.976	Int	0.970	0.086	11.279	0.001
			Slp	0.055	0.005	10.907	0.002
	50+	0.018	Int	1.546	0.154	10.024	0.002
			Slp	0.002	0.009	0.232	0.831

^1, 2^ Int = Intercept, Slp = Slope.

**Table 3 ijerph-20-03549-t003:** Multiple regression analysis (beta weights, standard errors, standardized coefficients, *t*, and *p*-values) of the contributions of pandemic-associated concerns to psychological distress (HADS).

	β	SE	Stand. Coeff.	*t*	*p*
Intercept	20.842	1.244		16.75	<0.001
Loneliness	0.936	0.111	0.202	8.40	<0.001
Chance	1.088	0.105	0.236	10.38	<0.001
Stability	−0.529	0.024	−0.564	−22.09	<0.001

**Table 4 ijerph-20-03549-t004:** Multiple regression analysis (beta weights, standard errors, standardized coefficients, *t*, and *p*-values) of the contributions of pandemic-associated concerns and Age to psychological distress (HADS).

	β	SE	Stand. Coeff.	*t*	*p*
Intercept	21.635	1.337		16.18	<0.001
Loneliness	0.950	0.112	0.205	8.49	<0.001
Chance	1.050	0.107	0.228	9.79	<0.001
Stability	−0.521	0.024	−0.556	−21.37	<0.001
Age	−0.020	0.013	−0.035	−1.57	0.118

## Data Availability

The NSDUH data are from the adult trend tables available at: https://www.samhsa.gov/data/sites/default/files/reports/rpt35323/NSDUHDetailedTabs2020v25/NSDUHDetailedTabs2020v25/NSDUHDetTabsSect10pe2020.htm (accessed on 26 July 2022). The HILDA data are taken from Figure 7.3 (p.113) in The Household, Income and Labour Dynamics in Australia Survey: Selected findings from Waves 1–19, The 16th Annual Statistical Report of the Hilda Survey, by R. Wilkins, E. Vera-Toscano, F. Botha, and S.C. Dahmann. The UKHLS data are publicly available for academic research purposes from the UK Data Service repository. The data from Myerson et al. [11], reanalyzed here in the section on Age and Individual Differences in Psychological Distress, are available along with fit statistics for new analyses of the data in Figures 1 and 3 at: https://osf.io/jmykd/?view_only=8df317eca67e4f2aa68848021c1a9a50 (accessed on 1 September 2022).

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
