# Peer review of "Distress Signals: Age Differences in Psychological Distress before and during the COVID-19 Pandemic"

_ijerph, 2023, doi:10.3390/ijerph20043549_

Round 1

Reviewer 1 Report (Previous Reviewer 1)

Congratulations to the authors for the work done.

Author Response

Reviewer 1:  Congratulations to the authors for the work done.

We are extremely pleased that this reviewer found this revision ready for publication.

Reviewer 2 Report (Previous Reviewer 2)

Thank you very much for the invitation to review the manuscript ijerph-2161857 - Distress Signals: Age Differences in Psychological Distress before and during the COVID-19 Pandemic.

Although the research object is very interesting, the text itself is confusing. The following factors lead me to think this way:

1. The organization / arrangement of subjects is not well connected. Although it is known that the public that reads the IJERPH is wide, it is expected that a text will minimally follow an organization capable of presenting the research object, the knowledge gap that it intends to "fill", the materials and methods that it used to arrive at the results and how the literature deals with this result previously. In this manuscript everything is very confusing... it starts with a quote that shouldn't even be there.

In addition there are a series of markings, which do not make sense

2. Methodological rigor: The authors do not state the study design. It's registered as a review, but that doesn't make any sense.

Review studies test hypotheses and aim to raise, gather, critically evaluate the research methodology and synthesize the results of several primary studies. Seeks to answer a clearly formulated research question. It uses systematic and explicit methods to retrieve, select and evaluate the results of relevant studies. Gathers and systematizes data from primary studies (units of analysis). It is considered the highest scientific evidence and is indicated in decision-making in clinical practice or public management.

3. It is not clear the quality of the data, based on the bases that originated them. Also, is it sensitive data, where is the approval by an ethics committee?

For these reasons, I am unable to indicate this text for publication.

Author Response

Reviewer 2:  Although the research object is very interesting, the text itself is confusing. The following factors lead me to think this way:

  1. The organization / arrangement of subjects is not well connected. Although it is known that the public that reads the IJERPH is wide, it is expected that a text will minimally follow an organization capable of presenting the research object, the knowledge gap that it intends to "fill", the materials and methods that it used to arrive at the results and how the literature deals with this result previously. In this manuscript everything is very confusing... it starts with a quote that shouldn't even be there.

In addition, there are a series of markings, which do not make sense

We are truly sorry that this reviewer had difficulties following our revised manuscript.  We had put considerable effort into dealing with their concerns regarding the organization, and even spelled out the logical order of the topics at the end of the Introduction (1st distress before the pandemic, 2nd distress during the pandemic, 3rd the determinants of individual distress levels, and 4th modeling how these differences affected the size of changes in distress) and then dealt with those topics sequentially.  We would attempt to do more, but the reviewer provides little guidance as to what they think would help.

  1. Methodological rigor: The authors do not state the study design. It's registered as a review, but that doesn't make any sense.

Review studies test hypotheses and aim to raise, gather, critically evaluate the research methodology and synthesize the results of several primary studies. Seeks to answer a clearly formulated research question. It uses systematic and explicit methods to retrieve, select and evaluate the results of relevant studies. Gathers and systematizes data from primary studies (units of analysis). It is considered the highest scientific evidence and is indicated in decision-making in clinical practice or public management.

It is true that the primary purpose of a review is often to evaluate the evidence, and indeed that had been our initial intention.  As we proceeded, however, we ended up following the science and writing a paper that, although it is best described as a review because no new data were collected, provides readers what they most desire from such an effort.  To begin with, there was so much agreement between the results of large, national studies using diverse “gold-standard” methodologies in different populations that a critique of the studies seemed unnecessary.  There was also so much agreement between studies of the effects of lockdowns that individual critiques did not seem called for.  In both cases, our novel contribution was to bring these results together so that readers can see that recent trends in psychological distress are truly international and to present analyses of individual-level data that lead to a new, explanatory model of these trends.  

  1. It is not clear the quality of the data, based on the bases that originated them. Also, is it sensitive data, where is the approval by an ethics committee?

It seems as if this reviewer is criticizing the NSDUH, HILDA, and UKHLS surveys, which would be strange without further information, although perhaps they are referring to our recently published study of distress and mitigation behaviors (Myerson et al., 2022).  In regard to the use of any sensitive data analyzed here, we would simply note that the data in all of our analyses of our own and others’ data had been previously deidentified.

For these reasons, I am unable to indicate this text for publication.

Reviewer 3 Report (New Reviewer)

The paper provides insightful evidence about age differences in psychological distress before and during the COVID-19 pandemic. The results add some meaningful contribution to literature. Anyway, there is some issue that needs to be handled. I encourage the authors to consider the comment given below and revise the paper accordingly in order to enhance the overall quality and completeness of the paper.

In the paper, the authors mentioned about age-related differences in personality traits which might provide some explanation why different age groups may encounter psychological distress differently during the covid-19 pandemic. However, there is no solid evidence to support this claim. In this point, it is important for the authors to provide a brief review of papers about some personality traits that were found to help people cope with the crisis situation from the covid-19 pandemic (such as mindfulness, psychological hardiness, resilience, optimism). In particular, I recommend the papers from the reputable journals below to be considered as the additional references for this review.

·       Effects of Trust in Organizations and Trait Mindfulness on Optimism and Perceived Stress of Flight Attendants during the COVID-19 Pandemic, Personnel Review. https://doi.org/10.1108/PR-06-2021-0396

·       How Does Mindfulness Help University Employees Cope with Emotional Exhaustion during the COVID-19 Crisis? The Mediating Role of Psychological Hardiness and the Moderating Effect of Workload, Scandinavian Journal of Psychology. 63(5), 449-461 https://doi.org/10.1111/sjop.12826

·       Cultivating Resilience During the COVID-19 Pandemic: A Socioecological Perspective. Annual Review of Psychology, 73(1), 575-598. https://doi.org/10.1146/annurev-psych-030221-031857

·       Psychological Resilience of Healthcare Professionals During COVID-19 Pandemic. Psychological Reports, 124(6), 2567-2586. https://doi.org/10.1177/0033294120965477

·       Big Five traits as predictors of perceived stressfulness of the COVID-19 pandemic. Personality and Individual Differences, 175, 110694. https://doi.org/https://doi.org/10.1016/j.paid.2021.110694

Author Response

Reviewer 3:  Comments and Suggestions for Authors

The paper provides insightful evidence about age differences in psychological distress before and during the COVID-19 pandemic. The results add some meaningful contribution to literature. Anyway, there is some issue that needs to be handled. I encourage the authors to consider the comment given below and revise the paper accordingly in order to enhance the overall quality and completeness of the paper.

In the paper, the authors mentioned about age-related differences in personality traits which might provide some explanation why different age groups may encounter psychological distress differently during the covid-19 pandemic. However, there is no solid evidence to support this claim. In this point, it is important for the authors to provide a brief review of papers about some personality traits that were found to help people cope with the crisis situation from the covid-19 pandemic (such as mindfulness, psychological hardiness, resilience, optimism). In particular, I recommend the papers from the reputable journals below to be considered as the additional references for this review.

We greatly appreciate this reviewer’s suggestions regarding relevant articles. Two stood out, and we now cite them in this revision.  The first is the paper by Zacher & Rudolph (2021, the last of the citations below) on Big 5 traits that provides valuable evidence regarding the association of Emotional Stability with psychological distress during the pandemic and that is consistent with the evidence from our own published study (Myerson et al., 2022) discussed in the section on Age and Individual Differences in Psychological Distress.  However, it seemed more important to cite it in the preceding section in reference to the effects of lockdowns on distress levels.  The reason stems from the fact that it was conducted in Germany, and thus directly addresses a concern that readers otherwise might have that the phenomena we are discussing all could be confined to English-speaking nations.   

Finally, we agree that a review of the literature on resilience was called for, and accordingly, although the literature is outside our area of expertise, we now refer readers to the timely review paper by Zhang et al. (2022, the third citation below) provided by this reviewer.  We also appreciated the other relevant articles cited by this reviewer and found them quite stimulating, but they seemed much more specific, and we suspect that readers would find the Zhang et al. review more helpful.

  • Effects of Trust in Organizations and Trait Mindfulness on Optimism and Perceived Stress of Flight Attendants during the COVID-19 Pandemic, Personnel Review. https://doi.org/10.1108/PR-06-2021-0396
  • How Does Mindfulness Help University Employees Cope with Emotional Exhaustion during the COVID-19 Crisis? The Mediating Role of Psychological Hardiness and the Moderating Effect of Workload, Scandinavian Journal of Psychology. 63(5), 449-461 https://doi.org/10.1111/sjop.12826
  • Cultivating Resilience During the COVID-19 Pandemic: A Socioecological Perspective. Annual Review of Psychology, 73(1), 575-598. https://doi.org/10.1146/annurev-psych-030221-031857
  • Psychological Resilience of Healthcare Professionals During COVID-19 Pandemic. Psychological Reports, 124(6), 2567-2586. https://doi.org/10.1177/0033294120965477
  • Big Five traits as predictors of perceived stressfulness of the COVID-19 pandemic. Personality and Individual Differences, 175, 110694. https://doi.org/https://doi.org/10.1016/j.paid.2021.110694

Round 2

Reviewer 2 Report (Previous Reviewer 2)

Thank you very much for returning the manuscript. Unfortunately, I still have difficulties understanding the organization of this paper, and I believe that this should be something that readers should also feel.

The organization of the manuscript has improved, but having two methods, two results... does not make sense and methodological details are still lacking.

Assuming that this is a secondary / revision text, what explains figures 1, or the regression table? This is data... how was this data managed? What about ethical permissions for involving data? if analyzes were made on data it is not a review... that makes no sense.

Author Response

This manuscript is a resubmission of an earlier submission. The following is a list of the peer review reports and author responses from that submission.

Round 1

Reviewer 1 Report

Comments to the author

Review on “Distress Signals: A Systematic Review of Psychological Distress before and during the COVID-19 Pandemic”

Thank you for the opportunity to review the manuscript.

Abstract: it presents the relevance of carrying out the present study, however, it does not allude to the type of study, nor to its objective.

The originality of the article is highlighted and evident, however the authors did not structure the presentation of their work, in my opinion, in an organized way, nor perceptible to the reader, making it difficult for other authors to reproduce the study. They do not present the methodology section, which is important in all investigations carried out. The authors present only the results and their discussion.

Reviewer 2 Report

There are many conceptual errors in various sections of the work that prevent publication.

1. Abstract: There are several excerpts that mention information that cannot be validated in a summary; or need references to something that cannot exist in an abstract;

The summary is incomplete; objectives, justification, methods and results are lacking. there is only discussion;

The text has primary formatting errors, words in different sizes, snippets that seem to have been pasted from other places;

This text cannot be called a review, as there is not even a method, it is impossible to understand the sources from which the data originate or even how they were processed;

The results are confusing due to the problem of their origin

The conclusion is huge and a mere repetition and/or continuation of the discussion

Reviewer 3 Report

1. Please improve the citation format for line 30-31 and all the tables' presentation

2. The method of this systematic review was unclear. There were no specific methods explaining the procedure/process of this systematic review. No details on the guidelines approach, studies assessment, studies protocol and criteria, data extraction and data analyses. I recommend changing this study type into a scoping review rather than a systematic review.

3. The flow of this reporting review was confusing between introduction, methods, results, and discussion. The authors should improve this flow so readers can understand their study